# Introducing Accurate 4-Bit Quantization with Hyperspherical Architecture

## Abstract

Due to the hardware support from NVIDIA's Blackwell architecture, 4-bit quantization of large language models promises substantial memory and throughput gains. However, naive 4-bit quantization degrades accuracy and remains challenging in practice. In this work, we revisit the root causes of this degradation and posit a new perspective through analysis of matrix multiplication and the unbounded weight within models. We show that quantization induces errors that are amplified within the attention and MLP submodules, leading to unstable error growth across layers. From this analysis, we propose architectural co-designs that use hyperspherical transformers to jointly normalize activations and constrain weights to unit norm, converting dot-products into bounded cosine similarities and suppressing error expansion. On 0.5–1B models, pretrained hyperspherical models yield new state-of-the-art performance to 4-bit weight-activation quantization, outperforming standard transformer architecture and a strong QAT baseline, while a partial normalization plug-in narrows the degradation gap in existing models. These results position model architectural co-design as a third optimization axis, complementary to existing works, for robust low-bit LLM deployment.

## 1 Introduction

Large Language Models (LLMs) have demonstrated unprecedented capabilities (Guo et al., 2025; Intelligence, 2024), but their deployment remains challenging due to the substantial memory and compute they require, making them impractical for many real-world applications, especially on resource-constrained devices (Qin et al., 2024; Zheng et al., 2025). Model quantization has emerged as a key strategy for lowering deployment costs, with recent efforts exploring 8-bit (Dettmers et al., 2022; Lin et al., 2024a) and even 4-bit (Liu et al., 2025; Ashkboos et al., 2024) representations for weights and activations. Such quantization enables reduced memory usage and faster computation, often while preserving acceptable service quality (Zhu et al., 2024; Czakó et al., 2025). Recent hardware roadmaps have also accelerated this trend by exposing native FP4 kernels with the NVIDIA Blackwell architecture, making 4-bit execution a realistic target rather than an academic curiosity.

However, plain FP4 quantization often causes severe generation quality degradation, even with more granular quantization schemas (Li et al., 2024). To mitigate this, prior works have focused on addressing activation and weight outliers during inference (Liu et al., 2025; Ashkboos et al., 2024; Chen et al., 2025; Kumar et al., 2024). Although these methods often restore quantized models to a usable state, notable empirical gaps in generation quality remain, along with the added complexity of model-specific adaptations that require special handling with some of the methods.

In this work, we dive into the fundamental reasons behind the accuracy degradation in 4-bit quantization. We reveal two important properties that were overlooked by previous works. First, quantization results in errors that are propagated and accumulated across layers. Second, the less-bounded weight rows in the models amplify the directional errors and cause unstable self-attention and MLP submodules within each layer. Motivated by this analysis, we explore hyperspherical transformers that normalize both weights and activations to unit norm (Loshchilov et al.). By converting dot-products into bounded cosine similarities, hyperspherical transformers are able to reduce error expansion in submodules and stabilize error propagation through depth (Luo et al., 2018). On three pretrained models ranging from 0.5–1B, hyperspherical architecture yields strong robustness at FP4 quantization, significantly outperforming standard transformer baselines.

**Our main contributions are:**

1. We characterize how quantization-induced errors interact with unbounded projections in attention/MLP submodules, yielding expanding and unstable propagation across layers.

2. Motivated by our analysis about unbounded projections, we introduce robust architectural co-design as a third axis to the quantization accuracy degradation mitigation techniques, distinct from existing post-training quantization (PTQ) techniques and quantization-aware training (QAT) frameworks.

3. We show that hyperspherical pretraining attains strong FP4 quantization robustness and makes significantly smaller accuracy degradation than (i) standard transformers and (ii) a strong QAT baseline. We also show that partial hypersphericity can narrow the quantization degradation equivalent to the previous state-of-the-art QAT framework with equal budget.

## 2 BACKGROUND AND RELATED WORK

### 2.1 QUANTIZATION

**Definition.** Quantization maps real-valued tensors to a low-precision discrete set to reduce memory, bandwidth, and compute cost. Given an input $x$, a quantization operator obtains an output $\hat{x}$ in reduced precision such that:

$$\hat{x} = \text{clip}(\lceil \frac{x}{s} \rfloor + z, q_{min}, q_{max}),$$

where $s > 0$ is the scaling factor, $z$ is the zero point (0 if the quantization method is symmetric), and $[q_{min}, q_{max}]$ bound the range after quantization. Scales can be chosen per-tensor, per-channel, per-group, or per-block (which is hardware-related). We will provide a more detailed discussion in Appendix A.2.1 on these scaling differences.

**Benefits of quantization.** Compared to a full precision model, a quantized model has two main benefits. First, with the reduced precision per parameter, it can be served on hardware with less memory, enabling inference of large models on edge devices (Lin et al., 2024a). Second, modern hardware supports native calculations in reduced precision, which are faster than full precision. Quantized models can leverage those kernels to provide faster inference (Gong et al., 2025).

**What to quantize?** Three components in an LLM affect the runtime memory consumption and efficiency: the weights of the model (W), intermediate activations that are produced during inference (A), and KV cache during the prefilling stage (KV) (Kwon et al., 2023). To achieve maximum reduction in runtime memory consumption, all three components shall be quantized (such as W8A8KV8), whereas only the weights and activations have a significant impact on the compute efficiency (such as W8A8) (Lin et al., 2024b; Xiao et al., 2023). Given that accurate quantization for W4A4 remains a significant challenge, our work focuses on enhancing efficiency and accuracy during inference, specifically targeting WA quantization in 4 bits (Liu et al., 2025; Xiao et al., 2023).

**Emergence of FP4.** Achieving practical inference efficiency requires co-design between quantization precision and hardware execution paths. Recent GPU architectures (e.g., NVIDIA Blackwell) expose native FP4 instructions and kernels. Accordingly, we adopt the vendor-specified FP4 encodings and consider FP4 quantization for both weights and activations (WA). Unless otherwise noted, we use micro-scaled block formats with a block size of 32 for MXFP4 and 16 for NVFP4, consistent with the corresponding hardware definitions.

### 2.2 EXISTING METHODS FOR ROBUST QUANTIZATION

The existing works in robust low-bit quantization can generally be categorized into two streams: Post-Training Quantization (PTQ) and Quantization-Aware Training (QAT).

**Post-Training Quantization (PTQ).** PTQ focuses on pushing weight and activation in LLMs to lower precisions with minimal or no fine-tuning. Classical techniques include per-channel (Jacob et al., 2018) and blockwise microscaling (Drumond et al., 2018) to control local dynamic range; improved calibration via clipping (Banner et al., 2018), percentile (Wu et al., 2020), entropy rules (Davoodi et al., 2019), and bias correction (Nagel et al., 2019); learned or Hessian/curvature-aware

rounding to directly minimize layer-wise reconstruction error (Nagel et al., 2020; Nahshan et al., 2021; Frantar et al., 2022); and handling activation outliers by migrating scale from activations to weights (e.g., activation smoothing) (Xiao et al., 2023; Yi et al., 2024) or by activation-aware weight selection (Lin et al., 2024a; Dettmers et al., 2022). State-of-the-art methods in this stream show strong performance, yet gaps persist at W4A4 precision (Ashkboos et al., 2024; Li & Panda, 2024; Lee et al., 2025).

**Quantization-Aware Training (QAT).** QAT integrates fake-quantization during pretraining or fine-tuning to simulate a quantized model and optimize model weights under a quantized environment (Chen et al., 2025). In practice, gradients through the non-differentiable quantizer are handled with the straight-through estimator (STE), where gradients are approximated and not automatically set through automatic differentiation during back propagation (Bengio et al., 2013). Beyond deterministic rounding, stochastic/learned rounding injects (or learns) zero-mean quantization noise during training to reduce bias and improve stability at low bit-widths (Zhao et al., 2024; Ozkara et al., 2025). QAT generally achieves the strongest accuracy at low bit-widths (including W4), at the cost of additional training (Liu et al., 2025).

This paper studies robust architectural co-design as a third direction that can potentially work with existing methods in parallel. Our analyses and results motivate a hyperspherical design that naturally supports reliable and accurate 4-bit quantization.

## 3 HYPERSPHERICITY LEADS TO QUANTIZATION ROBUSTNESS

In this section, we show that the hyperspherical model has more robustness to 4-bit quantization compared with regular transformer models. In particular, the unbounded weights in the submodules of transformer models result in unbounded variances in matrix multiplications, whereas normalization can effectively control and reduce the error variance through cosine similarity.

### 3.1 WHY UNBOUNDED PROJECTIONS AMPLIFY QUANTIZATION NOISE.

#### 3.1.1 INTUITIVE SCENARIO: A SMALL ERROR WITH A LARGE WEIGHT

We first start with a hypothetical example. As illustrated in Figure 1, consider one coordinate in the matrix multiplication $y_i = w_i^\top x$. If the $j$-th coordinate of $w_i$ is extremely large, say $|w_{ij}| = M \gg 1$, and the error $\varepsilon_j$ is small, then the output perturbation on that coordinate is $\Delta y_i = w_i^\top \varepsilon = \sum_k w_{ik}\varepsilon_k \approx M\varepsilon_j$. Even when $|\varepsilon|_2$ is tiny, $M\varepsilon_j$ can be large if $M$ is unbounded. **Due to unbounded weight, a small input error is amplified to a significant output error that ultimately affects the final generation.**

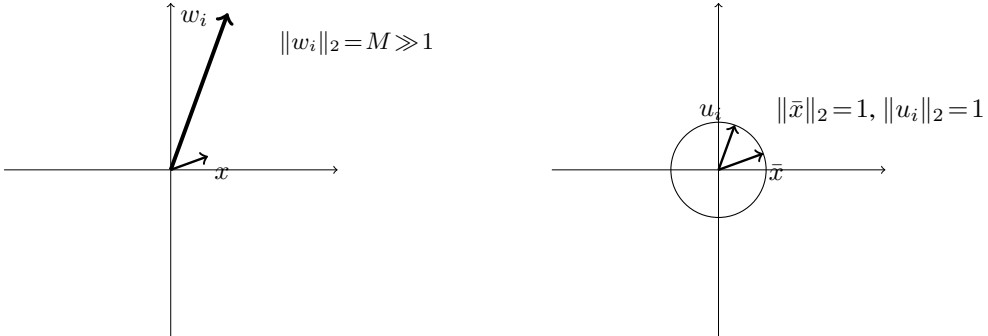

(a) **Standard projection** (dot product). Unconstrained $w_i$ allow *large* $\|w_i\|_2$, so small input errors can be multiplied by large weights.

(b) **Hypersphere projection** (cosine similarity). $\|\bar{x}\|_2 = 1$, $\|u_i\|_2 = 1$, and $0 \le g_i \le C$ convert the dot product into a *bounded* cosine projection.

Figure 1: Comparison of projection geometry and gain. An unconstrained dot product can exhibit unbounded local gains, amplifying quantization noise. Hyperspherical layers normalize the representation and bound projection directions and gains, capping amplification.

### 3.1.2 THEORETICAL DERIVATION

We formalize the intuition by modeling the quantization process as introducing noise to the weights and activations. We show that bounding both weights and activations is necessary for controlling the error induced by quantization.

**Activation-side variance bound.** Let $\tilde{x} = x + \varepsilon$ with $\mathbb{E}[\varepsilon] = 0$ and $\mathrm{Cov}(\varepsilon) = \Sigma_x \preceq \sigma^2 I$. Then $\Delta y = W\tilde{x} - Wx = W\varepsilon$.

**Proposition 1** (Activation-side variance).

$$
\mathbb{E}\big[\|\Delta y\|_2^2\big] = \mathrm{Tr}\big(W^\top W \Sigma_x\big) \ \leq \ \sigma^2\, \mathrm{Tr}(W^\top W) = \sigma^2\, \|W\|_F^2
$$
$$
\leq \ \sigma^2\, r\, \|W\|_2^2 \ \leq \ \sigma^2\, \min(d_{\mathrm{in}}, d_{\mathrm{out}})\, \|W\|_2^2, \tag{1}
$$

*where $r = Rank(W)$. Equivalently, for each output coordinate $i$, $\mathrm{Var}[\Delta y_i] = w_i^\top \Sigma_x\, w_i \leq \sigma^2 \|w_i\|_2^2$.*

*Proof.* By $\mathbb{E}[\varepsilon\varepsilon^\top] = \Sigma_x$ and cyclic trace, $\mathbb{E}\|\Delta y\|_2^2 = \mathrm{Tr}(W^\top W \Sigma_x)$. Because $\Sigma_x \preceq \sigma^2 I$ and $W^\top W \succeq 0$, $\mathrm{Tr}(W^\top W \Sigma_x) \leq \sigma^2\, \mathrm{Tr}(W^\top W) = \sigma^2 \|W\|_F^2$. Let $\{\sigma_k(W)\}_{k=1}^r$ be the nonzero singular values. Then $\|W\|_F^2 = \sum_{k=1}^r \sigma_k(W)^2 \leq r \max_k \sigma_k(W)^2 = r\, \|W\|_2^2$, which yields the final inequality in Equation 1. For the coordinate-wise claim, apply $\mathrm{Var}(w_i^\top \varepsilon) = w_i^\top \Sigma_x w_i \leq \sigma^2 \|w_i\|_2^2$. $\qquad\square$

**Relative amplification.** If $y = Wx \neq 0$, we have the relative amplification from matrix multiplication on the input as:

$$
\mathcal{A}_{\mathrm{act}}(W, x; \Sigma_x) = \frac{\mathrm{Tr}(W^\top W \Sigma_x)}{\|Wx\|_2^2} \ \leq \ \frac{\sigma^2\, \|W\|_F^2}{\|Wx\|_2^2}.
$$

When $\|W\|_F$ is large and $\|Wx\|_2$ happens to be small (e.g., $x$ aligns with a near-zero direction), $\mathcal{A}_{\mathrm{act}}$ can be large. This formalizes the instability risk without weight constraints (small errors have large impacts on the output).

**Weight-side variance bound.** Let $\tilde{W} = W + E$ with $\mathbb{E}[E] = 0$ and $\mathrm{Cov}(\mathrm{vec}(E)) \preceq \tau^2 I$. For fixed $x$, $\Delta y := \tilde{W}x - Wx = Ex$.

**Proposition 2** (Weight-side variance). *If $\|x\|_2 \leq R$ and entries $E_{ij}$ are independent with $\mathrm{Var}(E_{ij}) \leq \tau^2$, then*

$$
\mathbb{E}\big[\|\Delta y\|_2^2\big] = \mathbb{E}\,\|Ex\|_2^2 = \sum_{i=1}^{d_{out}} \sum_{j=1}^{d_{in}} x_j^2\, \mathrm{Var}(E_{ij}) \ \leq \ \tau^2 \|x\|_2^2\, d_{out} \ \leq \ \tau^2 R^2\, d_{out}.
$$

*Proof.* For each $i$, $\sum_j x_j E_{ij}$ is a sum of independent zero-mean variables. Thus $\mathrm{Var}[\sum_j x_j E_{ij}] = \sum_j x_j^2 \mathrm{Var}(E_{ij}) \leq \tau^2 \sum_j x_j^2$. Summing over $i$ yields the claim. $\qquad\square$

### 3.2 DEFINITION OF A HYPERSPHERICAL MODEL

A hyperspherical model constrains both activations and weights to live on (or close to) a unit hypersphere, so that linear projections reduce to cosine similarities with uniformly bounded gain.

Let $\mathcal{N} : \mathbb{R}^d \to \mathbb{R}^d$ denote L2 normalization, we have the normalization operator:

$$
\mathcal{N} \ = \ \frac{z}{\|z\|_2}.
$$

We also write row normalization of a matrix $W \in \mathbb{R}^{d_{\mathrm{out}} \times d_{\mathrm{in}}}$ as

$$
\mathrm{row\_unit}(W)_i \ = \ \frac{w_i}{\|w_i\|_2}.
$$

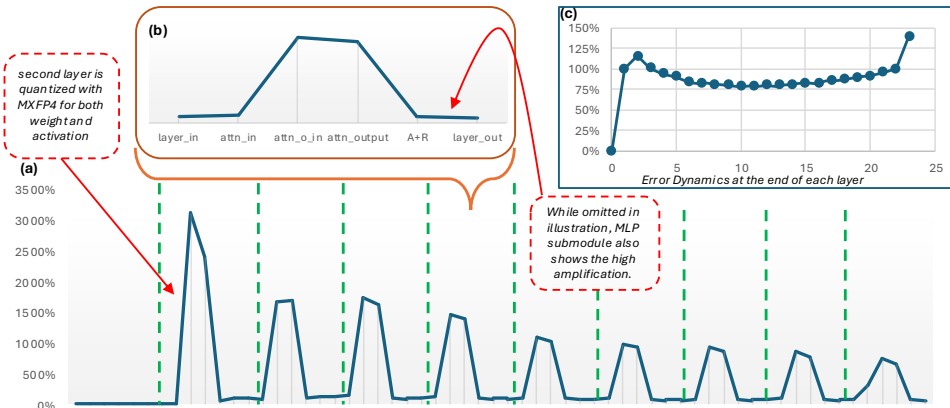

Figure 2: (a) Overview of error dynamics within a standard transformer's first ten layers after the second layer is quantized. (b) Error dynamics within one layer. We refer readers to Appendix A.4.1 for a visual mapping between labels on the horizontal axis and the positions in the block. (c) Error dynamics at the end of each layer when the second layer is quantized.

**Hyperspherical linear layer.** A hyperspherical linear layer replaces $y = Wx$ with

$$\bar{x} = \mathcal{N}(x), \qquad U = \text{row\_unit}(W), \qquad y = U\bar{x} \tag{2}$$

with each coordinate of $y$ satisfies

$$y_i = \langle u_i, \bar{x} \rangle = \cos\theta_i, \qquad \text{with } \|u_i\|_2 = 1, \ \|\bar{x}\|_2 = 1.$$

**Hyperspherical block.** For an attention/MLP block, we apply Equation 2 to all projections (e.g., $W_q, W_k, W_v, W_o, W_{\text{up}}, W_{\text{gate}}, W_{\text{down}}$). During the experiments, we chose to use an established implementation from Loshchilov et al.. Loshchilov et al. normalizes the activations and weights to a hypersphere representation through L2 normalization, with some added learnable parameters to aid in the model's convergence. In addition, Loshchilov et al. normalizes the embedding and lm_head weight, which we also adopt. The experiment setup is elaborated in Appendix A.3.2.

## 3.3 INSTABILITY IN TRANSFORMER VS. STABILITY IN HYPERSPHERICAL MODEL

To support the theoretical analysis, we reveal empirically that when a quantization-induced error is introduced in the input, its magnitude is expanded in the self-attention and MLP submodules in a standard transformer block. Figure 2 shows the error propagation pattern within each layer of a 0.5B transformer model (we omit the MLP submodule due to space, but it has a similar pattern to the attention submodule). It uses the normalized percentage of L2 magnitude as a proxy to quantify the existence of errors in the model. The percentage is calculated as $\frac{1}{|\mathcal{I}|} \sum_{i \in \mathcal{I}} \frac{\|\tilde{y}_i - y_i\|_2}{\|y_i\|_2}$, where $y_i$ and $\tilde{y}_i$ denote the full-precision and erroneous activations at index $i$ with $\mathcal{I}$ spanning all evaluated tokens in Wikitext2 dataset (Merity et al., 2016), and then normalized with respect to the percentage at the output of second layer. It should be evident that the relative magnitude of the input activation error of the layer is significantly amplified in the submodules.

This is consistent with our analyses in the previous subsection. RMSNorm limits *activation scale* but does not constrain the projection norms or directions of $W$. Proposition 1 therefore still scales with $\|W\|_F^2$, allowing large internal amplification.

Since quantization robustness is determined not only by where noise enters but also by the outputs of the projections it subsequently encounters, architectures that bound projection norms and operate on the normalized hypersphere representations are more robust to quantization (including FP4). Hypersphericity can bound the effective element-wise gain from each submodule, converting the unbounded multiplication into a bounded cosine similarity in a hypersphere. Concretely, we would observe such robustness through a reduction in the amplification of intermediate activations.

Empirically, the hyperspherical model exhibits significantly reduced amplification inside the attention submodules, as shown in Figure 3 (with the same visualization style as Figure 2). The internal error is no longer amplified by orders of magnitude, which in turn accelerates error attenuation across the layers and yields a smaller final error at depth.

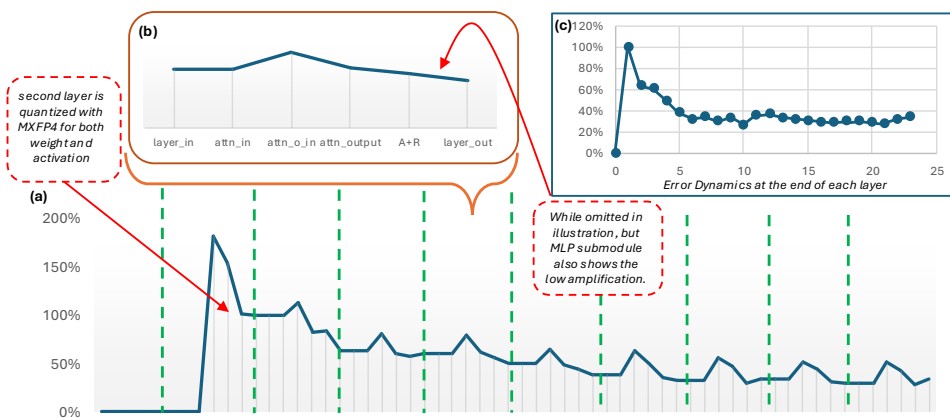

Figure 3: (a) Overview of error dynamics within a hyperspherical transformer's first ten layers after the second layer is quantized. (b) Error dynamics within one layer. We refer readers to Appendix A.4.1 for a visual mapping between labels on the horizontal axis and the positions in the block. (c) Error dynamics at the end of each layer when the second layer is quantized.

Table 1: Hyperspherical models' robustness generalizes across model sizes. T is the standard model architecture, whereas HS represents the hyperspherical architecture (Loshchilov et al.). The number in parentheses means relative accuracy degradation compared to full precision.

| Model | Precision | PIQA | HS | LAMB | Arc-E | SciQ | Average |
|---|---|---|---|---|---|---|---|
| T-0.5B | BF16 | 68.50% | 37.21% | 39.69% | 63.43% | 85.30% | 58.83% (−) |
| | MXFP4 | 65.45% | 34.21% | 20.67% | 54.71% | 77.20% | 50.45% (-14.24%) |
| | NVFP4 | 64.91% | 35.44% | 25.81% | 58.00% | 82.90% | 53.41% (-9.21%) |
| HS-0.5B | BF16 | 69.37% | 39.47% | 41.28% | 65.40% | 89.20% | 60.94% (−) |
| | MXFP4 | 67.95% | 38.55% | 36.70% | 63.17% | 88.00% | **58.87% (-3.40%)** |
| | NVFP4 | 68.23% | 39.07% | 39.12% | 64.31% | 88.80% | **59.91% (-1.69%)** |
| T-0.7B | BF16 | 70.29% | 39.00% | 41.55% | 65.66% | 85.80% | 60.46% (−) |
| | MXFP4 | 65.40% | 36.00% | 19.81% | 56.69% | 80.40% | 51.66% (-14.56%) |
| | NVFP4 | 68.01% | 37.44% | 32.25% | 62.12% | 83.80% | 56.72% (-6.19%) |
| HS-0.7B | BF16 | 70.35% | 40.37% | 41.26% | 67.63% | 88.10% | 61.54% (−) |
| | MXFP4 | 68.99% | 39.18% | 40.25% | 66.96% | 87.30% | **60.54% (-1.62%)** |
| | NVFP4 | 70.29% | 39.88% | 39.45% | 66.20% | 87.30% | **60.62% (-1.49%)** |
| T-1B | BF16 | 71.76% | 41.39% | 44.15% | 69.70% | 88.80% | 63.16% (−) |
| | MXFP4 | 68.88% | 37.42% | 26.31% | 61.57% | 84.50% | 55.74% (-11.75%) |
| | NVFP4 | 69.64% | 39.28% | 36.77% | 65.66% | 87.70% | 59.81% (-5.30%) |
| HS-1B | BF16 | 71.55% | 43.27% | 45.35% | 72.05% | 89.70% | 64.38% (−) |
| | MXFP4 | 70.95% | 42.14% | 40.13% | 69.28% | 89.90% | **62.48% (-2.95%)** |
| | NVFP4 | 70.13% | 42.96% | 43.12% | 71.00% | 89.20% | **63.28% (-1.71%)** |

### 3.4 HYPERSPHERICAL MODELS HAVE MUCH MORE ROBUST 4-BIT QUANTIZATION

We show that hyperspherical models with the above normalizations are much more accurate under 4-bit quantization compared with the standard transformer architecture. Table 1 shows the accuracy on five downstream tasks of three pairs of models of different sizes, where each pair was trained to a similar evaluation loss between a standard and a hyperspherical design (Loshchilov et al.). Hyperspherical models are clearly more robust to quantization than the standard architecture, with significantly less accuracy degradation. In Appendix A.4.2, we show that hyperspherical models can even perform better on downstream tasks compared to applying the latest QAT framework, ParetoQ (Liu et al., 2025), on the standard architecture.

### 3.5 PARTIAL HYPERSPHERICITY FOR EXISTING LLMs

We have shown in the previous subsection that the hyperspherical model architecture can lead to state-of-the-art quantization robustness. However, existing LLMs are largely non-weight-bounding architectures, and it is a non-trivial task to retrain a hyperspherical model from scratch. In this

Table 2: INT4 (weight-only) quantization for 0.5B model. QAT is adopted from Liu et al. (2025).

| Method | Time | PPL | PIQA | HS | LAMB | Arc-E | SciQ | **Average** |
|--------|------|-----|------|-----|------|-------|------|---------|
| BF16 | – | 13.04 | 68.50% | 37.21% | 39.69% | 63.43% | 85.30% | 58.83% |
| – | – | 22.12 | 66.16% | 36.19% | 15.08% | 55.13% | 78.40% | 50.19% |
| QAT | 12:06:57 | 14.49 | **68.34%** | 36.42% | 36.10% | **61.66%** | 83.10% | 57.12% |
| v1 | 11:07:37 | 16.20 | 67.52% | **36.61%** | 34.58% | 58.29% | 80.50% | 55.50% |
| v1+QAT | 12:05:59 | 14.48 | 68.23% | 36.41% | **37.53%** | 60.52% | 82.30% | 57.00% |
| v2 | 11:03:06 | **13.82** | 68.06% | 36.51% | 36.46% | 61.15% | **84.10%** | **57.26%** |
| v2+QAT | 12:01:14 | 14.82 | 66.81% | 34.90% | 28.29% | 59.39% | 81.70% | 54.22% |

Table 3: INT4 (weight-only) quantization for 1B model. QAT is adopted from Liu et al. (2025).

| Method | Time | PPL | PIQA | HS | LAMB | Arc-E | SciQ | **Average** |
|--------|------|-----|------|-----|------|-------|------|---------|
| BF16 | – | 11.53 | 71.76% | 41.39% | 44.15% | 69.70% | 88.80% | 63.16% |
| – | – | 25.06 | 68.39% | 38.15% | 23.29% | 58.29% | 80.30% | 53.68% |
| QAT | 23:41:54 | **12.49** | 71.49% | **41.14%** | 37.38% | 67.80% | 86.90% | 60.94% |
| v1 | 22:00:13 | nan | **71.87%** | 41.03% | **43.68%** | 67.17% | 87.20% | **62.19%** |
| v1+QAT | 23:42:17 | nan | 70.78% | 40.96% | 41.96% | 67.47% | 89.00% | 62.03% |
| v2 | 21:49:33 | 12.61 | 70.29% | 40.28% | 42.11% | **68.27%** | **88.60%** | 61.91% |
| v2+QAT | 23:38:36 | 21.24 | 64.47% | 32.63% | 22.88% | 56.90% | 82.50% | 51.88% |

section, we present our attempts to make existing models more robust to 4-bit quantization through partial hysphericity.

We consider two levels of architectural changes based on two characteristics of the error analyses. First, weight normalization is essential to control the output variances and stability of submodules. However, exploratory experiments have revealed that normalization of all weights will cause the model to fall back to an untrained state, even when the original model's weights are migrated. Thus, we take a step back and only convert the final projection (*lm_head*) into a hyperspherical architecture. This choice is motivated by two reasons. Firstly, the input of *lm_head* retains all features from previous layers. Leaving the previous layers' architecture unchanged would make the model converge much faster during finetuning. Secondly, *lm_head* is the last layer, and variances of its output directly impact the output of the model. In the results, we will denote this approach as **v1**.

Second, input magnitude to the submodules effectively controls the output magnitude. Knowing that the submodules in a standard model are not stable, we can reduce the input strength of those submodules with stronger normalization. Concretely, **in addition to v1**, we replace the RMSNorm with L2Norm by removing the learnable gain and the $\frac{1}{\sqrt{d}}$ factor in the denominator. This approach makes the error propagation factor closer to 1 by taking fewer effect from the submodules. Ideally, such an approach would result in a more stable error propagation, but we note that more changes to the architecture will lead to more difficulties in convergence when finetuning the model with a limited budget. In the results, we will denote this approach as **v2**.

We apply QAT and our normalization approaches to the pretrained 0.5B and 1B standard transformer models and summarize their quantization robustness after finetuning in Table 2 and 3. In the table, a dash indicates a naive quantization without any finetuning, and QAT is from Liu et al. (2025). We report the perplexity on Wikitext2 and five downstream tasks, as well as the finetuning wall time (Time). We choose INT4 weight-only quantization (W4) to be consistent with Liu et al. (2025)'s experiments. We also evaluate FP4 quantization strategies, as well as the Pythia family models (Biderman et al., 2023). Due to space limits, we provide those additional results in Appendix A.4.3. Partial hyphericity (v1) and heavier regularization on input in submodules (v2) can perform on par (and sometimes slightly better) than QAT while using around 8% less time, which we attribute to the QAT framework utilizing specifically designed backpropagation. In contrast, architectural changes are fully compatible with the optimized backpropagation library in PyTorch.

## 4   UNDERSTANDING ACCURACY DEGRADATION IN 4BIT QUANTIZATION

In this section, we discuss practices for obtaining accurate FP4 quantized models by synthesizing previous understandings of the causes of accuracy degradation in 4-bit quantization and providing

new insights through the lens of error propagation. While successfully reproducing some phenomena (Kumar et al., 2024; An et al.; Sun et al., 2024), our analyses reveal new findings, in particular suggesting that outliers in LLM are not consistently the cause of accuracy degradation, with an emphasis on error analysis. The experiment settings can be found in Appendix A.3.3.

### 4.1 LESS IS MORE IN PRETRAINING

First, we extend the previous discussion on the relationship between prolonged pretraining and the model's accuracy after quantization. Kumar et al. (2024) has found that whereas the full precision model's accuracy keeps improving with more pretraining tokens, INT3-quantized models suffer from worse accuracy after pretraining for a certain number of tokens. We similarly observe this for the 4-bit quantization schema. As shown in Figure 4, the relative perplexity after MXFP4 WA quantization grows with more training tokens from the start, and optimal perplexity after MXFP4 quantization emerges around pretraining with 40B tokens. In addition to reproducing this effect, we find that maintaining a high learning rate might delay the critical point, but the convergence is hard to reach with a high constant learning rate. We discuss this in Appendix A.5.1.

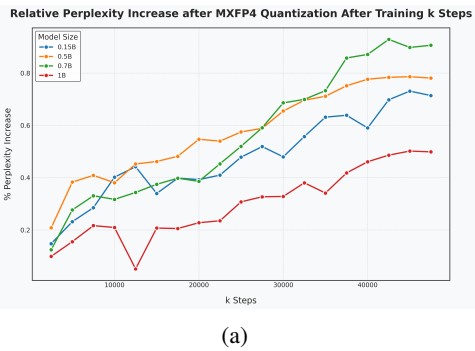 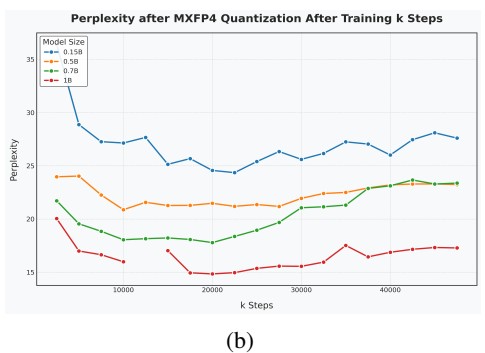

| (a) | (b) |

Figure 4: (a) As training goes on, accuracy degradation after MXFP4 quantization as measured by percentage increase of perplexity on Wikitext2 (Merity et al., 2016) dataset. (b) As training goes on, perplexity after quantization stops decreasing and starts to increase. Discontinuity of the 1B model is an outlier with a perplexity value of 1301.

### 4.2 OUTLIERS MATTER, BUT ONLY TO AN EXTENT

Many previous works propose methods for eliminating the outliers that emerge in the inference stage of LLMs as a mitigation to the accuracy degradation (Dettmers et al., 2022; Ashkboos et al., 2024). Although outliers intuitively influence the quantization range and can suppress other activation values to zero during quantization, their impact on final accuracy under blockwise scaling has been underemphasized in previous works. Our findings confirm the presence of these outliers and show that they contribute to some, but not all, of the accuracy degradation in quantization. Due to space limitations, we provide more detailed discussions in Appendix A.5.2.

### 4.3 ERROR PROPAGATION DRIVES THE ANALYSIS

In Section 3, we have already shown that unbounded weights in the standard transformer model lead to unstable and expanding error propagation in submodules. In this section, the error analysis is generalized to show that quantization error has a non-monotonic layer-wise growth while being divergent in nature. We present some connections between error and existing understandings of outliers, and hope our perspective of thinking from error propagation can motivate the community and be informative in designing future model architectures.

Without loss of generalizability, we consider a transformer model where the l-th layer is defined as:

$$\text{Layer}_l(x_l) = x_l + \text{Attn}(\text{RMSNorm}(x_l)) + \text{MLP}(\text{RMSNorm}(x_l + \text{Attn}(\text{RMSNorm}(x_l)))) \quad (3)$$

For simplicity of analysis, we abstract the computation in the layer with a function $f$, where $x_{l+1} = x_l + f_l(x_l)$. The effect of quantization on the previous layer can be modelled as producing an erroneous output ($\tilde{x}$) and introducing an error term $e_l = \tilde{x}_l - x_l$. In this section, we consider the

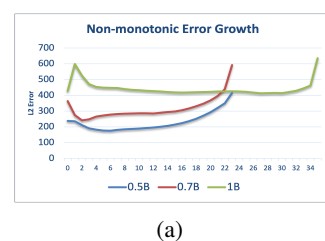 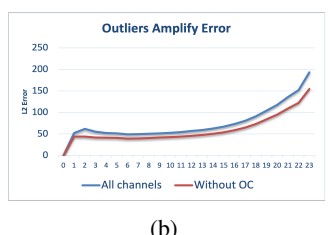 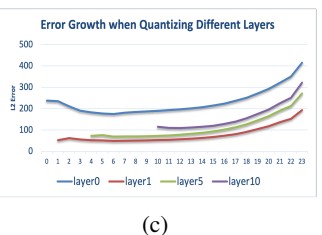

(a)                 (b)                 (c)

Figure 5: (a) Visualization of L2 error propagation measured at the end of each layer, after quantizing the first layer in the three models. (b) Outliers amplify the initial error and affect the subsequent propagated errors. (c) When quantizing exactly one layer at a time, the best-to-worst ranking of four choices is Layers [1, 5, 10, 0], which corresponds perfectly to the L2 magnitude of the error.

first-order approximation of error propagation. Denote the Jacobian of $f_l$ at $x_l$ by $J_l = Df_l(x_l)$, where $D$ is the Frechet derivative. On Layer $l$, we have:

$$\tilde{x_{l+1}} = x_l + e_l + f_l(x_l + e_l) \simeq x_l + e_l + f_l(x_l) + J_l e_l \tag{4}$$

Assuming the current layer is not quantized (i.e., no independent error is initiated from this layer), the error propagated to the next layer will be in the form of: $e_{l+1} = e_l + J_l e_l = (I + J_l)e_l$. As the layers deepen, the error propagation follows a recursive pattern, where at the end of layer $k$, an error term that originates from layer $l$ will become:

$$e_k^{(l)} = \prod_{i=l+1}^{k} (I + J_i)e_l \tag{5}$$

**Non-monotonic Layer-wise Growth** Equation 5 focuses on the final error propagation of the error from the l-th layer, and we can generalize this pattern to all layers that are quantized. For a set of layers that is quantized $Q \subseteq \{0, 1, ...N-1\}$, we have the final error at the end of N layers to be:

$$E^{final} = \sum_{l \in Q} \prod_{i=l+1}^{N} (I + J_i)e_l \tag{6}$$

Note that since $||I + J_i||$ is not guaranteed to be greater than 1, the layer-wise error growth is non-monotonic. A simple bound over the error propagation amplification factor is $1 - ||J_i|| < ||I + J_i|| < 1 + ||J_i||$. We will provide a discussion on when the error will diverge in Appendix A.6.1 for the general Jacobian matrix and in Appendix A.6.2 for a model-specific Jacobian decomposition. Empirically, Figure 5a illustrates the growth of error that is introduced by quantizing the first layer of three models of different sizes.

**Outliers Amplify Initial Error.** The existence of outliers will amplify the error when quantizing a specific layer, as shown in Figure 5b. This is the reason why outliers negatively affect quantization. We also note that in practice, since all layers are quantized and outliers (especially OCs) exist in all layers, the effect of OCs is worsened because every layer's input error is amplified.

**Implications from Error Propagation.** Understanding error propagation aids in comprehending the accuracy degradation that occurs during quantization. It can also serve as a proxy for determining which layers to skip quantization under mixed-quantization frameworks: less error magnitude means less accuracy degradation during quantization. As a simple demonstration, when quantizing exactly one layer at a time, the best-to-worst ranking of four choices is Layers [1, 5, 10, 0]. This ordering mirrors the relative error propagation that initiates at those same layers, as shown in Figure 5c.

## 5 CONCLUSION

In this paper, we show that accuracy degradation from quantization stems from the error propagated across layers and amplified inside attention/MLP submodules. Based on our analysis, we then reveal that bounding both activations and weights on a hypersphere stabilizes and reduces propagation. Our analysis and experiments show that hyperspherical transformers are much more robust against FP4 quantization, and a lightweight normalization plug-in on the LM head is able to narrow the gap on existing models and match a strong QAT method. Our analysis and experiments posit robust architecture designs to be a practical third dimension for robust 4-bit LLMs.

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

## A   APPENDIX

### A.1   STATEMENTS

#### A.1.1   STATEMENT ON THE USAGE OF LLM

Large language models were utilized in the paper to refine writing and correct grammatical errors, as well as generating an initial version of Figure 1b. The authors take full responsibility for the content of this manuscript.

#### A.1.2   STATEMENT ON ETHICS

This work and the authors adhere to the ICLR Code of Ethics. In this work, no human or animal subjects were involved. All datasets are properly cited and sourced in compliance with their usage guidelines. No harmful information, bias, or discriminatory content exists as a result of our research, to the best of our knowledge. We do not need to declare additional conflicts of interest and sponsorship beyond those that will be filtered by the OpenReview system.

#### A.1.3   STATEMENT ON REPRODUCIBILITY

As of the time of this submission, the authors are uncertain whether the code associated with this work can be released to the public due to company policies. For reproducibility, we kindly refer readers to the implementation provided by Loshchilov et al. for the model details. Important hyperparameters are disclosed in Appendix A.3. Experiments of quantization are implemented with standard round-to-nearest symmetric blockwise quantization. All experiments are run with seed 42.

If the reviewers/readers have any difficulties in reproducing the work, they are advised to raise questions (during the anonymity period) or contact the authors (after the anonymity period).

### A.2   EXTRA BACKGROUND AND RELATED WORKS

#### A.2.1   DISCUSSION ON THE SCALING FACTOR

Figure 6 illustrates how different scaling methods are applied during the quantization. The input activations are a batch of 2D matrices, and the weights in the model are also typically 2D matrices. Per-tensor scaling calculates a single scaling factor for the entire 2D matrix, row-wise/column-wise scaling applies a separate scaling factor for each row or column, and group-wise scaling uses a separate scaling factor per group. We want to explicitly differentiate group-wise scaling and block-wise scaling, with the latter one being used in our FP4 quantization methods. Group-wise and block-wise scaling are functionally the same, where they apply one scaling factor per some pre-defined number of parameters ($S$). However, block-wise scaling is a hardware-specific scaler, where $S$ is decided by the input dimension of the reduced-precision computation kernel. For example, the MXFP4 kernel takes $S = 32$, whereas the NVFP4 kernel uses $S = 16$. On the other hand, group-wise scaling does not restrict $S$. Thus, increasing $S$ can lead to less memory consumption of the model, but the efficiency is the same unless customized kernels are developed.

### A.3   EXPERIMENT SETUPS

#### A.3.1   GENERAL HARDWARE/SOFTWARE SETUP.

All experiments are done on either NVIDIA H200 or A100 GPU clusters. Each cluster is equipped with 8 GPUs. Evaluation of the models is with a single-card setting, and pretraining/finetuning of the models uses either one cluster of H200 or multiple clusters of A100. All GPUs use the CUDA 12.8 driver. All models are in HuggingFace format (Wolf et al., 2019), with Pytorch 2.7.0 (Paszke et al., 2019) and Transformers 4.52.0. Training/model hyperparameters differ for different settings and will be reported separately below for each section.

All quantizations are zero-centered symmetric quantizations with the simple round-to-nearest method. For INT4 weight quantization, we use llm-compressor (AI & vLLM Project, 2024) and apply column-wise scaling. For MXFP4 and NVFP4 quantization, since we do not have access

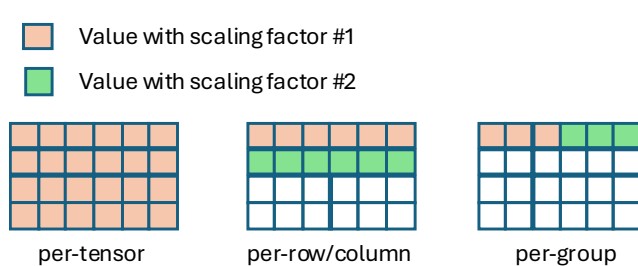

Figure 6: Illustration of different scaling methods in quantization.

to Blackwell GPUs, we implement our own fake quantization framework, where the values are first quantized to the respective precision and immediately dequantized. Per kernel specifications, MXFP4 uses a block size of 32 and NVFP4 uses a block size of 16 for both weights and activations.

This paper evaluates models on one benchmark for perplexity and five common benchmarks for downstream tasks. We use Wikitext2 Merity et al. (2016) to evaluate perplexity, and PIQA (Bisk et al., 2020), HellaSwag (Zellers et al., 2019), LAMBADA-openai Paperno et al. (2016), Arc-easy Clark et al. (2018), SciQ (Welbl et al., 2017), to evaluate downstream task ability.

### A.3.2 HYPERPARAMETER FOR SECTION 3

Section 3 involves both pre-training and finetuning. Standard transformer models are pre-trained with the same configuration as in Appendix A.3.3. Hyperspherical models are trained with the same configuration as their respective standard models, with differences in warmup steps and weight decays removed, which are also used by Loshchilov et al.. We take the pre-trained Pythia checkpoints from Huggingface (Biderman et al., 2023).

Finetuning is done on roughly 10B tokens from the Fineweb dataset (Penedo et al., 2024), processed in the respective tokenizer. We tested a few learning rates and chose the best one in favor of the QAT model. The final determined learning rate was 6e-4 for the standard architecture and 5e-5 for Pythia. All finetuning uses cosine decay to 0 without weight decay and warmup steps, which is consistent with Liu et al. (2025). All figures shown in this section are from MXFP4 quantization. Tables will separately specify their quantization method.

### A.3.3 HYPERPARAMETER FOR SECTION 4

Section 4 involves the analysis of four standard architecture models of different sizes. Models are trained with the general setup. The 0.15B model was an exploratory model, which we did not use for formal studies. The three large ones (0.5B, 0.7B, 1B) are pretrained with roughly 100B tokens from the Fineweb dataset (Penedo et al., 2024), processed using a public tokenizer. We use a learning rate of 4e-3 with 2000 steps of warmup and cosine decay to 0. The global batch size is 1000 with gradient accumulation being 1. Weight decay is 0.1. Maximum context length is 2048. The 0.5B and 0.7B models are 24 layers, and the 1B model is 36 layers. All figures shown in this section are from MXFP4 quantization.

### A.4 ADDITIONAL EXPERIMENTS FOR SECTION 3

### A.4.1 SKETCH FOR SUB-LAYER MEASUREMENT POSITIONS IN A STANDARD TRANSFORMER'S BLOCK

Figure 7 annotates the positions of measurement of the normalized percentage of L2 magnitude for Figure 2 and 3.

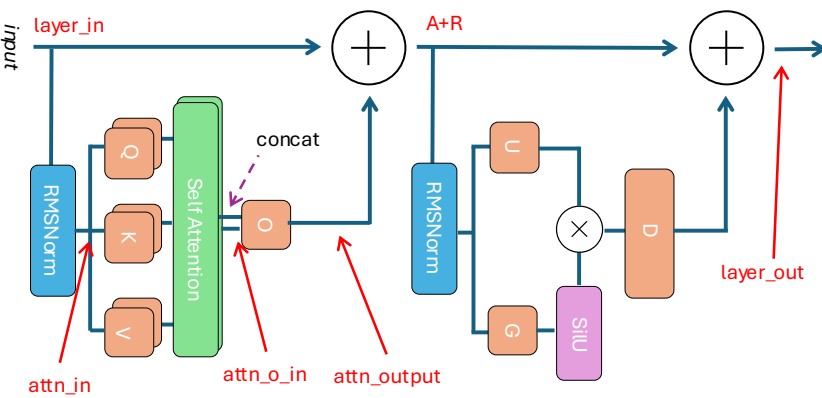

Figure 7: Red arrows and notations mark where the L2 percentage errors are measured within each block.

Table 4: Hyper-spherical models are more robust to quantization compared to standard architectures, even accompanied with QAT. The number in parentheses means relative accuracy degradation compared to full precision.

| Model | Precision | Arc-E | HS | LAMB | PIQA | SciQ | Average |
|---|---|---|---|---|---|---|---|
| T-0.5B | BF16 | 68.50% | 37.21% | 39.69% | 63.43% | 85.30% | 58.83% (–) |
| | MXFP4 | 65.45% | 34.21% | 20.67% | 54.71% | 77.20% | 50.45% (-14.24%) |
| | NVFP4 | 64.91% | 35.44% | 25.81% | 58.00% | 82.90% | 53.41% (-9.21%) |
| QAT-0.5B | MXFP4 | 67.14% | 34.60% | 31.07% | 57.58% | 85.20% | 55.12% (-6.31%) |
| | NVFP4 | 67.63% | 36.10% | 31.01% | 60.98% | 82.60% | 55.66% (-5.39%) |
| HS-0.5B | BF16 | 69.37% | 39.47% | 41.28% | 65.40% | 89.20% | 60.94% (–) |
| | MXFP4 | 67.95% | 38.55% | 36.70% | 63.17% | 88.00% | **58.87% (-3.40%)** |
| | NVFP4 | 68.23% | 39.07% | 39.12% | 64.31% | 88.80% | **59.91% (-1.69%)** |
| T-1B | BF16 | 71.76% | 41.39% | 44.15% | 69.70% | 88.80% | 63.16% (–) |
| | MXFP4 | 68.88% | 37.42% | 26.31% | 61.57% | 84.50% | 55.74% (-11.75%) |
| | NVFP4 | 69.64% | 39.28% | 36.77% | 65.66% | 87.70% | 59.81% (-5.30%) |
| QAT-1B | MXFP4 | 68.39% | 38.78% | 36.48% | 65.99% | 88.80% | 59.69% (-5.49%) |
| | NVFP4 | 70.89% | 40.40% | 35.94% | 67.93% | 87.50% | 60.53% (-4.16%) |
| HS-1B | BF16 | 71.55% | 43.27% | 45.35% | 72.05% | 89.70% | 64.38% (–) |
| | MXFP4 | 70.95% | 42.14% | 40.13% | 69.28% | 89.90% | **62.48% (-2.95%)** |
| | NVFP4 | 70.13% | 42.96% | 43.12% | 71.00% | 89.20% | **63.28% (-1.71%)** |

### A.4.2 ADDITIONAL TABLE FOR SECTION 3.4

Here, we provide additional support that hyperspherical models are even more robust than standard models finetuned with QAT. We apply the latest QAT framework, ParetoQ (Liu et al., 2025), on the pretrained 0.5B and 1B models, and summarize the comparisons in Table 4. It is evident that the hyper-spherical models are more robust to quantization compared to QAT models on the standard architecture.

### A.4.3 ADDITIONAL TABLE FOR SECTION 3.5

As aforementioned in Section 3.5, we provide additional results for quantizing partial-normalization (v1 and v2) and QAT models into FP4 precision for both weight and activations. We also provide results showing the generalization of the proposed methods on Pythia models. Tables 5 to 8 present the quantized model's performance on the downstream tasks under the two different FP4 specifications for 0.5B and 1B models. Table 9 and 10 show quantizing Pythia to INT4 after QAT vs. our

Table 5: MXFP4 quantization for 0.5B model. QAT is adopted from Liu et al. (2025).

| Method | Time | PPL | PIQA | HS | LAMB | Arc-E | SciQ | **Average** |
|--------|------|-----|------|-----|------|-------|------|---------|
| BF16 | – | 13.04 | 68.50% | 37.21% | 39.69% | 63.43% | 85.30% | 58.83% |
| – | – | 23.23 | 65.45% | 34.21% | 20.67% | 54.71% | 77.20% | 50.45% |
| QAT | 12:06:57 | 16.68 | 67.14% | 34.60% | 31.07% | 57.58% | **85.20%** | 55.12% |
| v1 | 11:07:37 | 19.42 | 65.56% | 34.48% | 22.55% | 55.89% | 82.80% | 52.26% |
| v1+QAT | 12:05:59 | 16.08 | 66.00% | 34.66% | 30.74% | **59.34%** | 84.80% | 55.11% |
| v2 | 11:03:06 | **15.58** | **67.36%** | **35.39%** | **32.21%** | 58.16% | 83.80% | **55.38%** |
| v2+QAT | 12:01:14 | 16.65 | 65.51% | 33.92% | 24.14% | 54.38% | 81.60% | 51.91% |

Table 6: MXFP4 quantization for 1B model. QAT is adopted from Liu et al. (2025).

| Method | Time | PPL | PIQA | HS | LAMB | Arc-E | SciQ | **Average** |
|--------|------|-----|------|-----|------|-------|------|---------|
| BF16 | – | 11.53 | 71.76% | 41.39% | 44.15% | 69.70% | 88.80% | 63.16% |
| – | – | 17.31 | 68.88% | 37.42% | 26.31% | 61.57% | 84.50% | 55.74% |
| QAT | 23:41:54 | **13.33** | 68.39% | 38.78% | 36.48% | 65.99% | 88.80% | **59.69%** |
| v1 | 22:00:13 | 14.36 | 68.44% | 38.31% | 36.25% | **66.33%** | 88.40% | 59.55% |
| v1+QAT | 23:42:17 | 13.36 | 68.28% | 38.72% | 35.67% | 65.78% | **89.10%** | 59.51% |
| v2 | 21:49:33 | 13.59 | **68.93%** | **39.15%** | **36.50%** | 65.74% | 88.10% | 59.68% |
| v2+QAT | 23:38:36 | 27.35 | 61.15% | 31.50% | 15.85% | 51.18% | 79.10% | 47.76% |

approaches. The v1 method has comparable and slightly better robustness compared to Liu et al. (2025).

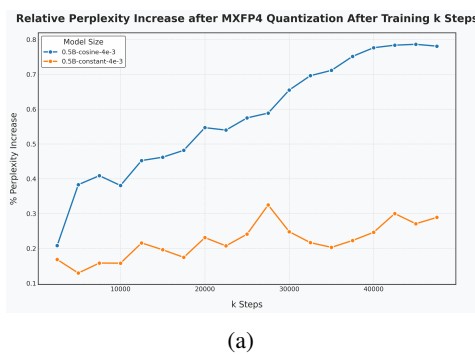
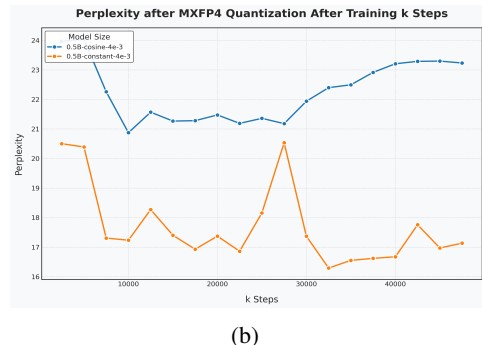

(a)                 (b)

Figure 8: (a) As training goes on, accuracy degradation after quantization measured by percentage difference of perplexity on Wikitext2 (Merity et al., 2016) dataset increases. (b) As training goes on, perplexity after quantization stops decreasing and starts to increase.

Table 7: NVFP4 quantization for 0.5B model. QAT is adopted from Liu et al. (2025).

| Method | Time | PPL | PIQA | HS | LAMB | Arc-E | SciQ | **Average** |
|---|---|---|---|---|---|---|---|---|
| BF16 | – | 13.04 | 68.50% | 37.21% | 39.69% | 63.43% | 85.30% | 58.83% |
| – | – | 16.93 | 65.18% | 35.79% | 26.16% | 58.16% | 82.70% | 53.60% |
| QAT | 12:06:57 | 15.54 | 67.63% | **36.10%** | 31.01% | **60.98%** | 82.60% | 55.66% |
| v1 | 11:07:37 | 15.27 | 67.08% | 35.77% | 35.46% | 59.85% | **83.90%** | 56.41% |
| v1+QAT | 12:05:59 | 14.95 | 67.79% | 35.98% | **35.65%** | 60.35% | 82.80% | **56.51%** |
| v2 | 11:03:06 | **14.51** | **68.12%** | 36.06% | 34.39% | 60.94% | 82.70% | 56.44% |
| v2+QAT | 12:01:14 | 15.74 | 65.72% | 34.73% | 26.84% | 57.41% | 81.00% | 53.14% |

Table 8: NVFP4 quantization for 1B model. QAT is adopted from Liu et al. (2025).

| Method | Time | PPL | PIQA | HS | LAMB | Arc-E | SciQ | **Average** |
|---|---|---|---|---|---|---|---|---|
| BF16 | – | 11.53 | 71.76% | 41.39% | 44.15% | 69.70% | 88.80% | 63.16% |
| – | – | 13.83 | 70.02% | 39.35% | 36.56% | 65.32% | 87.70% | 59.79% |
| QAT | 23:41:54 | 12.72 | 70.89% | 40.40% | 35.94% | **67.93%** | 87.50% | 60.53% |
| v1 | 22:00:13 | 12.77 | **71.06%** | 40.08% | 38.48% | 67.34% | **88.60%** | **61.11%** |
| v1+QAT | 23:42:17 | **12.69** | 70.13% | **40.59%** | 38.44% | 67.76% | 88.10% | 61.00% |
| v2 | 21:49:33 | 12.89 | **69.75%** | 39.96% | **39.41%** | 66.79% | 88.30% | 60.84% |
| v2+QAT | 23:38:36 | 24.33 | 63.71% | 32.03% | 20.16% | 54.97% | 82.50% | 50.67% |

Table 9: INT4 quantization for Pythia-410M model. QAT is adopted from Liu et al. (2025).

| Method | PIQA | HS | LAMB | Arc-E | SciQ | **Average** |
|---|---|---|---|---|---|---|
| BF16 | 66.70% | 33.73% | 51.64% | 51.89% | 81.50% | 57.09% |
| – | 64.47% | 32.28% | 23.73% | 46.04% | 72.40% | 47.78% |
| QAT | 65.94% | **34.39%** | 29.07% | 56.14% | 80.80% | 53.27% |
| v1 | 65.89% | 33.79% | **34.85%** | 56.02% | **83.60%** | **54.83%** |
| v1+QAT | **65.94%** | 33.88% | 32.41% | **56.90%** | 81.80% | 54.19% |
| v2 | 56.15% | 27.08% | 6.85% | 41.25% | 67.70% | 39.81% |
| v2+QAT | 53.37% | 25.69% | 0.12% | 27.53% | 26.50% | 26.64% |

Table 10: INT4 quantization for Pythia-1B model. QAT is adopted from Liu et al. (2025).

| Method | PIQA | HS | LAMB | Arc-E | SciQ | **Average** |
|---|---|---|---|---|---|---|
| BF16 | 70.73% | 37.78% | 56.28% | 56.99% | 83.90% | 61.14% |
| – | 67.25% | 36.01% | 49.06% | 52.95% | 84.40% | 57.93% |
| QAT | 67.74% | **37.27%** | 41.63% | 60.19% | 84.60% | 58.29% |
| v1 | **68.50%** | 37.26% | 40.85% | **61.87%** | **86.50%** | **59.00%** |
| v1+QAT | 68.12% | 36.65% | **42.73%** | 59.47% | 85.60% | 58.51% |
| v2 | 59.19% | 27.60% | 6.33% | 45.75% | 69.20% | 41.61% |
| v2+QAT | 57.78% | 27.14% | 9.76% | 40.82% | 70.40% | 41.18% |

## A.5 ADDITIONAL EXPERIMENTS FOR SECTION 4

### A.5.1 THE EFFECT OF LEARNING RATE DECAY ON QUANTIZATION

The analysis in the main text is done with cosine learning rate decay with a minimum learning rate of 0. We conduct ablation studies on keeping the learning rate constant on a 0.5B model, trained with other parameters being the same. The results are shown in Figure 8a. With a constant learning rate of 4e-3, the accuracy degradation from 4-bit quantization seems less compared to cosine decay. However, we still see a trend of percentage perplexity increasing as training goes on, hinting at an eventual emergence of the critical point with a higher token-parameter ratio.

### A.5.2 DISCUSSION ON ACTIVATION OUTLIERS DURING INFERENCE

**Definition of Outliers** We follow the previous works (An et al.; He et al., 2024; Sun et al., 2024; Raman et al., 2025) in categorizing the outliers. An et al. discusses both weight and activation outliers. In this analysis, we focus on activation outliers, which are much larger in magnitude. The prevailing view in the community is that there are two types of activation outliers in the LLM: outlier channels (OCs) and massive activations (MAs). An outlier channel is defined as a channel in the activation where the mean exceeds the overall average of the tensor by more than $m$ standard deviations with a standard deviation less than $\beta$ (Raman et al., 2025). On the other hand, a massive activation is a single value in the activation with an extremely high value, often in the multiple thousands, compared to others Sun et al. (2024). In simple words, an OC is an entire channel where each element has a large value, whereas an MA is a large activation value corresponding to a single token. An illustrative example of OCs and MAs is provided in Figure 9a and 9b.

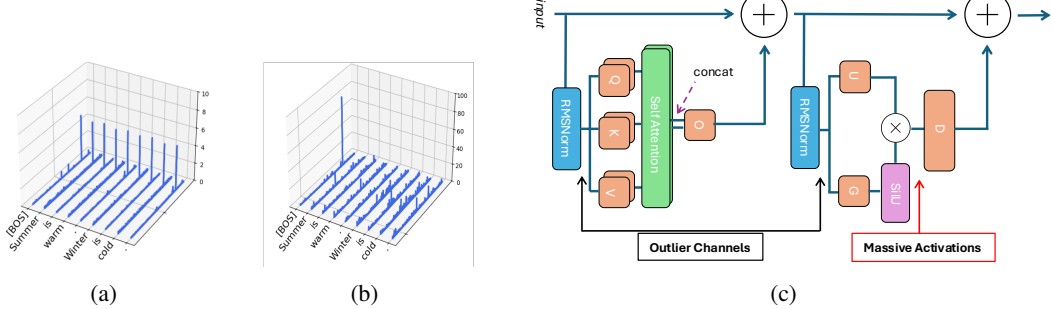

| (a) | (b) | (c) |

Figure 9: (a) Outlier Channels. (b) Massive Activation. (c) Where are the outliers?

**Where are the outliers?** Interestingly, outlier channels and massive activations consistently appear in different places within each layer of the LLMs. As shown in Figure 9c, for the standard transformer architecture, OCs emerge in the input activations of QKV projections in the attention module and in the input activations of up/gate projection in the MLP module, and massive activations appear in the input activations of the down projection. A more detailed examination of those outliers reveals that OCs are composed of two parts: the carried-over OC through residual connections and the emerging OC from the current layer's computation. When these two components agree on the channel index, the OC's magnitude will become larger and larger, and when these two components do not agree on the index, OCs might change in the later layers. Compared to OCs that appear on all tokens and consistent channels, Sun et al. (2024) first found that MAs usually appear on some random channels in the first token or semantically weak tokens. Our observations are in line with that finding.

**How much do the outliers impact accuracy?** In our investigation of 4-bit WA quantization, we find that the effect of outliers on accuracy degradation is significantly influenced by the granularity of the quantization method employed. While previous research has often attributed accuracy degradation primarily to activation outliers, our analysis reveals a more nuanced relationship. When using row/column-wise scaling for weights and activations, activation outliers indeed account for a substantial portion of the accuracy degradation. However, our experiments demonstrate that weight and other non-outlier activations also play a non-trivial role in the overall accuracy degradation and should not be ignored when aiming at minimizing the impact of 4-bit quantization. This finding chal-

lenges the conventional wisdom that addressing activation outliers alone is sufficient for maintaining model accuracy. In particular, group quantization emerges as an effective strategy for mitigating the impact of outlier channels (OC). When the outliers could not affect the entire activation tensor, their impact is much less. Figure 10a illustrates the impact of outliers on activations during quantization. The negative impacts of outliers are only constrained to their neighbors in the block.

To systematically evaluate the effect of outliers on the final accuracy after quantization, we conducted a hierarchical analysis comparing three quantization scenarios: standard weight and activation quantization (all channels), weight and activation quantization with outlier channels untouched (w/o OC), and weight-only quantization (no channels). The results in Figure 10b demonstrate that 1. while OCs do impact quantization performance as indicated by the differences between the orange and yellow bars, this impact is not uniform across different model sizes, and 2. quantization error from weights can lead to at least 20% perplexity increase on the Wikitext2 dataset (Merity et al., 2016). In addition, we empirically observe that MAs pose fewer degradations compared to OCs on our models. MAs contribute to around 10% among all perplexity increases caused by activation quantization in our 0.5B model, whereas OC contributes to around 50%. However, this relationship may vary with model scale, as MAs are reportedly more hazardous in larger models (Raman et al., 2025).

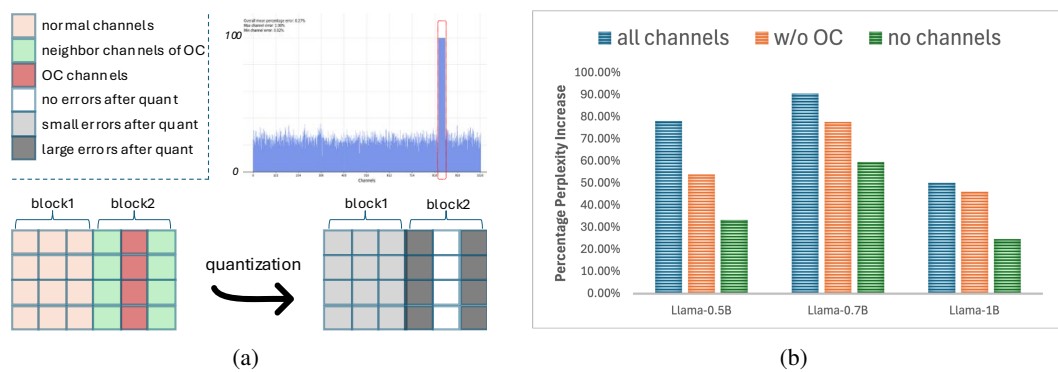

(a)                    (b)

Figure 10: (a) OCs only affect their neighbor channels in the same block. The top right figure shows the quantization error measured by the percentage of error over the original magnitude. Values affected by OC (channels in the red dotted square) have a high error percentage, but others have a lower percentage. (b) Percentage of perplexity increase after quantization of weight and activation (all channels), weight and activation without outliers (w/o OC), weight only (no channels).

## A.6 DISCUSSION ON THE JACOBIAN MATRIX

### A.6.1 GENERAL DISCUSSION ON THE ERROR DIVERGENCE AND JACOBIAN

We first discuss the divergence of the error when the Jacobian matrix is generic (not specific to any architecture). Recall that under the first-order approximation, we have

$$e_{l+1} = e_l + J_l e_l = (I + J_l)e_l \tag{7}$$

For simplicity, we assume the Jacobian matrix is independent of the error with a random distribution, and we have:

$$\begin{aligned} \mu &= \mathbb{E}(J_l) \\ \Sigma &= \mathbb{E}((J_l - \mu)^T (J_l - \mu)) \end{aligned} \tag{8}$$

We have the squared norm of $e_{l+1}$ to be:

$$\begin{aligned} \|e_{l+1}\|^2 &= e_l^T (I + J_l)^T (I + J_l)e_l \\ &= e_l^T (I + J_l + J_l^T + J_l^T J_l)e_l \end{aligned} \tag{9}$$

We then have the conditional expectation with respect to $e_l$ as:

$$\begin{aligned} \mathbb{E}(\|e_{l+1}\|^2 | e_l) &= e_l^T \mathbb{E}(I + J_l + J_l^T + J_l^T J_l)e_l \\ &= e_l^T (I + \mu^T + \mu + \mu^T \mu + \Sigma)e_l \end{aligned} \tag{10}$$

We denote $(I + \mu^T + \mu + \mu^T \mu + \Sigma)$ as $M$, and using Rayleigh quotient bound, we have:

$$e_l^T M e_l \geq \lambda_{min}(M)\|e_l\|^2, \tag{11}$$

which implies that $\mathbb{E}(\|e_N\|^2) \geq \prod_{k=0}^{N-1} \lambda_{min}(M_k)\mathbb{E}(\|e_0\|^2)$

**Implication from Equation 11.** Here, we discuss the conditions under which the error will diverge or converge.

- if $J$ is zero-centered, $M = I + \Sigma \succcurlyeq I$. Consequently, the mean-square error is non-decreasing. The error will diverge when the geometric mean of $\prod_{k=0}^{N-1} \lambda_{min}(M_k)$ is greater than 1.
- in a similar derivation, $\mathbb{E}(\|e_N\|^2) \leq \prod_{k=0}^{N-1} \lambda_{max}(M_k)\mathbb{E}(\|e_0\|^2)$, which means the error will converge when the geometric mean of $\prod_{k=0}^{N-1} \lambda_{max}(M_k)$ is smaller than 1.
- we note that under the typical setting, with i.i.d., zero-mean random Jacobians with variance $\Sigma \succcurlyeq 0$, the first-order error will almost certainly diverge.

### A.6.2 MODEL SPECIFIC JACOBIAN AND ITS ERROR GROWTH BOUND

In this section, we will further decompose the Jacobian of the standard architecture and show the upper and lower bounds for error propagation in that architecture. First, recall from Eq. 3 that the architecture is made of an attention module and an MLP module. Let $x \in \mathbb{R}^{T \times d}$ be the input matrix to a transformer block and let

$$A(x) := \text{Attn}(\text{RMSNorm}(x)), \qquad f(x) := x + A(x) + \text{MLP}(\text{RMSNorm}(x + A(x))).$$

We study the Jacobian $J(x) := Df(x)$ as a linear map acting on matrix perturbations (error introduced in the input) $\epsilon \in \mathbb{R}^{T \times d}$. All bounds are given in the Frobenius norm $\|\cdot\|_F$ on matrices together with the induced operator norm:

$$\|\mathcal{T}\|_{F \to F} := \sup_{\epsilon \neq 0} \frac{\|\mathcal{T}[\epsilon]\|_F}{\|\epsilon\|_F} \qquad \text{for any linear map } \mathcal{T} : \mathbb{R}^{T \times d} \to \mathbb{R}^{T \times d}.$$

We define the following notations at a base point $x$,

$$z = \text{RMSNorm}(x), \quad u = x + A(x), \quad w = \text{RMSNorm}(u).$$

By the chain rule,

$$J(x) = I + B(x) + M(x)(I + B(x)), \qquad \begin{cases} B(x) := J_{\text{Attn}}(z) J_{\text{RMS}}(x), \\ M(x) := J_{\text{MLP}}(w) J_{\text{RMS}}(u). \end{cases} \tag{12}$$

**Row-wise RMSNorm and its Jacobian.** RMSNorm is a *row-wise* transformation. For row $t \in \{1, \ldots, T\}$, let $x_t \in \mathbb{R}^d$ be the $t$-th row of $x$ and

$$\sigma_t = \sqrt{\tfrac{1}{d}\|x_t\|_2^2}, \qquad D_g = \mathrm{diag}(g) \in \mathbb{R}^{d \times d} \text{ (learned gains).}$$

Then,

$$\mathrm{RMSNorm}(x)_t = D_g \frac{x_t}{\sigma_t}$$

$$J_{\mathrm{RMS}}(x) = \mathrm{blkdiag}\Big(N_1, \ldots, N_T\Big)$$

with the per-row Jacobian matrix being:

$$N_t = \frac{D_g}{\sigma_t}\left(I - \frac{x_t x_t^\top}{d\,\sigma_t^2}\right) \in \mathbb{R}^{d \times d}. \tag{13}$$

Consequently,

$$\|J_{\mathrm{RMS}}(x)\|_{F \to F} = \max_t \|N_t\|_2 \le \frac{\|D_g\|_2}{\min_t \sigma_t}. \tag{14}$$

We'd like to make a side note on the error propagation after RMSNorm: after normalization, radial errors disappear in the input, but tangential errors remain.

**Blockwise bounds for $B$ and $M$.** For notation simplicity, we define the Frobenius-induced Lipschitz moduli:

$$L_{\mathrm{attn}}(z) := \|J_{\mathrm{Attn}}(z)\|_{F \to F}, \qquad L_{\mathrm{mlp}}(w) := \|J_{\mathrm{MLP}}(w)\|_{F \to F}.$$

By submultiplicativity and Eq. 14, we have the bounds for $\|B\|$ and $\|M\|$ as:

$$\|B(x)\|_{F \to F} \le L_{\mathrm{attn}}(z)\,\|J_{\mathrm{RMS}}(x)\|_{F \to F} \le L_{\mathrm{attn}}(z)\frac{\|D_g\|_2}{\min_t \sigma_t}, \tag{15}$$

$$\|M(x)\|_{F \to F} \le L_{\mathrm{mlp}}(w)\,\|J_{\mathrm{RMS}}(u)\|_{F \to F} \le L_{\mathrm{mlp}}(w)\frac{\|D_g'\|_2}{\min_t \sqrt{\tfrac{1}{d}\|u_t\|_2^2}}, \tag{16}$$

where $u_t$ is the $t$-th row of $u$. [1]

**Per-layer error amplification bounds.** For any perturbation $\epsilon \in \mathbb{R}^{T \times d}$,

$$J(x)\epsilon = \epsilon + B(x)\epsilon + M(x)\epsilon + M(x)B(x)\epsilon$$

Using the triangle inequality and submultiplicativity, we have the upper bound for the error propagation as follows:

$$\boxed{\|J(x)\epsilon\|_F \le \Big(1 + \|B(x)\|_{F \to F} + \|M(x)\|_{F \to F} + \|M(x)\|_{F \to F}\|B(x)\|_{F \to F}\Big)\|\epsilon\|_F} \tag{17}$$

Likewise, we have the lower bound of error propagation as:

$$\boxed{\|J(x)\epsilon\|_F \ge \Big(1 - \|B(x)\|_{F \to F} - \|M(x)\|_{F \to F} - \|M(x)\|_{F \to F}\|B(x)\|_{F \to F}\Big)\|\epsilon\|_F,} \tag{18}$$

which is informative whenever

$$\|B(x)\|_{F \to F} + \|M(x)\|_{F \to F} + \|M(x)\|_{F \to F}\|B(x)\|_{F \to F} < 1$$

---

[1] RMSNorms are different for attention and MLP modules, so we use a different notation $D_g'$ for MLP

**Additional discussions on $J_{\text{Attn}}$ and $J_{\text{MLP}}$.** The exact analytical forms of $J_{\text{Attn}}$ and $J_{\text{MLP}}$ are overly complex and do not have much meaningful contributions to the error propagation in realistic LLMs. However, if we were to make some assumptions to loosen the constraints, we may obtain coarse bounds for them that contribute to the architectural changes.

We make the following three assumptions:

1. RMSNorm can bring normalized activations to $O(1)$.

2. The operator norm of the softmax Jacobian is smaller than some constant $c_{sm} < 1$.

3. Norms of SiLU in MLP are small. We have $\|\text{SiLU}\|_\infty < c_{\text{SiLU}}$ and $\|\text{SiLU}'\|_\infty < c_{\text{SiLU}'}$.

Under the three assumptions, we have a coarse single-head attention bound as:

$$\|J_{\text{Attn}}(z)\|_{F \to F} \lesssim \|W_o\|_2 \Big( \|W_v\|_2 + c_{\text{sm}} \frac{\|W_q\|_2 \|W_k\|_2}{\sqrt{d_k}} \Big)$$

We can also extend this to the practical multi-head attention case:

$$\|J_{\text{Attn}}(z)\|_{F \to F} \lesssim \|W_o\|_2 \sqrt{H} \max_h \Big( \|W_v^{(h)}\|_2 + c_{\text{sm}} \frac{\|W_q^{(h)}\|_2 \|W_k^{(h)}\|_2}{\sqrt{d_k}} \Big),$$

where $H$ is the number of heads.

Similarly, we have the following coarse bound for $J_{\text{MLP}}$ as:

$$\|J_{\text{MLP}}(w)\|_{F \to F} \lesssim \|W_d\|_2 \Big( c_{\text{SiLU}} \|W_u\|_2 + c_{\text{SiLU}'} \|W_g\|_2 \Big)$$

**Implications from the above analysis.** From the mathematical derivations, we make the following observations. Some of them can help us understand how to make architectural changes to better reduce the error propagation in LLMs.

- The residual connection guarantees a baseline error propagation factor of 1. Submodules add $B + M + MB$ to the next layer.

- Since RMSNorm eliminates radial errors, $B + M + MB$ contains mostly tangential (directional) errors that can either grow or shrink depending on the alignment between input and weight.

- Since weight norms are unbounded, the output error can have very high magnitudes if the alignment is not good.

- Some directions to reduce error propagation (or at least keep the error propagation factor around one) include: reducing weight magnitude, keeping the RMSNorm's gain $D_g$ small, reducing the number of heads in attention, and encouraging $B\epsilon$ and $M(I + B)\epsilon$ to point at the opposite direction of $\epsilon$.

