# OpenReview forum: "Introducing Accurate 4-Bit Quantization with Hyperspherical Architecture"
_ICLR.cc/2026/Conference — ICLR 2026 Conference Withdrawn Submission_

### Official Review · Reviewer_di54 · 2025-10-21

**Soundness:** 2
**Presentation:** 3
**Contribution:** 2
**Rating:** 2
**Confidence:** 4

**Summary:**

In this work, a hyperspherical architecture has been introduced to enhance the performance after quantization.

The key idea is to normalize weight matrices (row-wise) and activations, thereby constraining the outputs to be located within the unit-norm circle.

From the experimental results under FP4 / INT4 quantization settings, the efficacy of the proposed architecture has been demonstrated.

**Strengths:**

- The idea to restrict the output power within a unit norm seems to be a reasonable method for the performance enhancement.
- I checked the proofs in the main context (I did not check for the proofs in Appendices), and they seem to be correct and provide some insight about the quantization error propagation.

**Weaknesses:**

- The most crucial concern is that the proposed concept has only been validated on small-scale LLMs (up to 1B). The validation on larger models and recent models known to be difficult to be quantized (e.g., Llama3-8B) is needed.
- I doubt whether the proposed hyperspherical architecture is efficient, since we need to conduct the row-wise normalization for every weight matrix and normalization for every activation. This will incur additional processing time during the actual inference. I also think that further validation is needed to show that the proposed architecture does not limit the training performance for large-scale models.
- Could the authors provide the performance of v1 under FP4 quantization? While the performance of full hyperspherical architecture has been measured under MXFP4, the performance of partial hyperspherical architecture has been examined under INT4. Please conduct the experiments under the same setting to see the importance of row-wise weight normalization.
- While the authors claim that their research provides the third direction for quantization, synergy with existing methods has not been investigated. Is the proposed method compatible with existing outlier suppression techniques (e.g., QuaRot, SpinQuant, OSTQuant)? Is the proposed method compatible with existing weight-quantization techniques (e.g., GPTQ, BoA, GPTAQ)?
- I recommend the authors explain the basic knowledge required for FP4 quantization. While most experimental results are obtained under FP4 quantization, most of preliminary contents are related to the integer quantization, which makes readers who are not familiar with FP quantization difficult to understand the paper.

**Questions:**

See Weaknesses.

- Which LLM model family has been used? I might miss the details, but it seems that the authors reported only model sizes.
- Which weight quantizer was used to obtain the results in Tables 2 and 3 (v1, v2, not v1+QAT, v2+QAT)?

---

### Official Review · Reviewer_AaNM · 2025-10-28

**Soundness:** 2
**Presentation:** 2
**Contribution:** 2
**Rating:** 2
**Confidence:** 5

**Summary:**

The paper observes that the unbounded projection in standard Transformer architectures leads to quantization error accumulation across layers, amplifying performance degradation after low-bit quantization.
To address this, the authors introduce a normalization scheme that constrains activation and weight magnitudes, aiming to stabilize the model and reduce quantization noise propagation.
They further provide a mathematical analysis showing that bounding the weights can minimize the quantization error, and present empirical results suggesting that the proposed hyperspherical model mitigates accuracy loss after INT4 quantization.

**Strengths:**

- The paper clearly identifies and articulates the connection between unbounded representations and error amplification in low-bit quantization.

- The proposed normalization-based approach is intuitive and aligns with recent trends toward geometric or norm-constrained representations (e.g., hyperspherical parameterizations).

- The empirical validation demonstrates that constraining activations and weights can indeed improve quantization stability and model robustness.

**Weaknesses:**

- The contribution appears incremental. Similar normalization-driven stabilization has been extensively explored in prior works such as nGPT, Peri-LN, and especially BitNet, which integrates normalization and clipping for ternary/binary quantized networks. The paper does not clearly differentiate its design or novelty from these existing approaches, and a more detailed structural and numerical comparison with BitNet in particular is needed to clarify the distinct contributions of this work.

-No comprehensive comparison with recent INT4 quantization methods (e.g., outlier smoothing, range calibration, or mixed-precision PTQ/QAT techniques) is provided, and it remains unclear how much additional gain the proposed architectural modification offers compared to purely algorithmic approaches.

**Questions:**

- An analysis of the hardware cost introduced by the additional normalization layers seems necessary.

- Given that prior architectures applying normalization for model stabilization (e.g., Peri-LN, nGPTQ) would likely not suffer significant degradation after quantization, how does the proposed method compare against such normalized baselines in terms of both performance and efficiency?

**Details Of Ethics Concerns:**

.

---

### Official Review · Reviewer_eA5R · 2025-10-31

**Soundness:** 2
**Presentation:** 3
**Contribution:** 3
**Rating:** 2
**Confidence:** 4

**Summary:**

1. This paper focuses on solving accuracy degradation in 4-bit LLM quantization.
2. It identifies that unbounded weights in standard transformers amplify quantization errors and cause unstable cross-layer propagation.
3. The authors propose a hyperspherical transformer (normalizing weights/activations to unit norm) and partial hypersphericity plugins.
4. Experiments on 0.5–1B models show hyperspherical designs outperform standard transformers and QAT baselines.
5. This work establishes architectural co-design as a third optimization axis for low-bit LLM deployment.

**Strengths:**

This article proposes a new quantitative paradigm, which I believe is very innovative and worthy of in-depth research in the future. The design of the hypersphere can indeed avoid outliers, which is worthy of further exploration and investigation.

**Weaknesses:**

1. There are also very efficient methods in existing PTQ frameworks; for example, QuaRot adopts the computational invariance of SliceGPT. The paper does not compare efficiency with such methods.
2. v2 replaces RMSNorm with LayerNorm, but there is no analysis of the impact of this change on model performance.
3. v1 only modifies the lm_head, but the main weight quantization content of LLM lies in the numerous linear layers within the decoder block, and the paper does not mention how to perform hypersphere transformation on them.
4. In terms of the performance of the quantized model, the effectiveness of the method proposed in the paper is not high. Rotation methods in PTQ (QuaRot, OSTQuant) can basically achieve an accuracy loss of less than 1%.
5. Experimental results for W4A4 are missing.
6. Quantization model size is limited, i.e., less than 1B. Current quantization works often apply 7-70B scales.

**Questions:**

How to extend the method to KV cache quantization?

---

### Official Review · Reviewer_pApf · 2025-10-31

**Soundness:** 3
**Presentation:** 2
**Contribution:** 2
**Rating:** 4
**Confidence:** 3

**Summary:**

Authors show that quantization induces errors that are amplified within the attention and MLP submodules, leading to unstable error growth across layers. They propose architectural co-designs
that use hyperspherical transformers to jointly normalize activations and constrain
weights to unit norm, converting dot-products into bounded cosine similarities and
suppressing error expansion.
On 0.5–1B models, pretrained hyperspherical models yield new state-of-the-art performance to 4-bit weight-activation quantization,
outperforming standard transformer architecture and a strong QAT baseline.

**Strengths:**

Table 1 Shows that Hyperspherical models’ robustness generalizes across model sizes.

**Weaknesses:**

Experiments are done on models with size up to 1B parameters.
It is good start, but evaluation on 8B model would be more valuable.

Table 2 shows INT4 (weight-only) quantization for 0.5B model. But results are mixed, there is no clear winner: some time "QAT" is better, some time "v1" is better, some time "v1 + QAT" is better.

**Questions:**

What is the overhead of applying normalization on activation and weights?

---

### Note · Authors · 2025-11-17

**Comment:**

Most reviewers want to see the model size scale up to 8B. We'd like to emphasize that to enjoy the benefits of hyperspherical architecture fully, the model needs to be **pretrained**. Unfortunately, we do not have the resources to do so for 8B models.

**Withdrawal Confirmation:**

I have read and agree with the venue's withdrawal policy on behalf of myself and my co-authors.